# Genome-Wide Association Study Reveals *CLEC7A* and *PROM1* as Potential Regulators of *Paracoccidioides brasiliensis*-Induction of Cytokine Production in Peripheral Blood Mononuclear Cells

**DOI:** 10.3390/jof9040428

**Published:** 2023-03-30

**Authors:** Ana Marina B. de Figueiredo, Jéssica Cristina dos Santos, Brenda Kischkel, Edwin Ardiansyah, Marije Oosting, Grazzielle Guimarães Matos, Iara Barreto Neves Oliveira, Frank van de Veerdonk, Mihai G. Netea, Célia Maria de Almeida Soares, Fátima Ribeiro-Dias, Leo A. B. Joosten

**Affiliations:** 1Laboratório de Imunidade Natural (LIN), Instituto de Patologia Tropical e Saúde Pública, Universidade Federal de Goiás, Goiânia 74605-050, Goiás, Brazil; 2Department of Internal Medicine, Radboud Institute for Molecular Sciences (RILMS) and Radboud Center of Infectious Diseases (RCI), Radboud University Medical Center, 6524 Nijmegen, The Netherlands; 3Departamento de Microbiologia, Instituto de Ciência Biomédicas, Universidade de São Paulo, São Paulo 05508-000, São Paulo, Brazil; 4Laboratório de Biologia Molecular, Instituto de Ciências Biológicas, Universidade Federal de Goiás, Goiânia 74690-900, Goiás, Brazil

**Keywords:** Paracoccidioidomycosis, GWAS, genetic polymorphism, *CLEC7A*, *PROM1*, CD38

## Abstract

Paracoccidioidomycosis (PCM) is a systemic mycosis caused by fungi of the genus *Paracoccidioides* and the different clinical forms of the disease are associated with the host immune responses. Quantitative trait loci mapping analysis was performed to assess genetic variants associated with mononuclear-cells-derived cytokines induced by *P. brasiliensis* on 158 individuals. We identified the rs11053595 SNP, which is present in the *CLEC7A* gene (encodes the Dectin-1 receptor) and the rs62290169 SNP located in the *PROM1* gene (encodes CD133) associated with the production of IL-1β and IL-22, respectively. Functionally, the blockade of the dectin-1 receptor abolished the IL-1β production in *P. brasiliensis-*stimulated PBMCs. Moreover, the rs62290169-GG genotype was associated with higher frequency of CD38^+^ Th1 cells in PBMCs cultured with *P. brasiliensis* yeasts. Therefore, our research indicates that the *CLEC7A* and *PROM1* genes are important for the cytokine response induced by *P. brasiliensis* and may influence the Paracoccidioidomycosis disease outcome.

## 1. Introduction

Paracoccidioidomycosis (PCM) is a systemic fungal infection endemic in Latin America, where South America is responsible for the most cases of the disease [1]. The disease is caused by the thermodimorphic fungi of the genus *Paracoccidioides.* The first species was described as *Paracoccidioides brasiliensis* [2] and this species has a wide distribution in Latin America, from Mexico to Argentina [3]. 

Humans infected with *Paracoccidioides* spp. may be asymptomatic or manifest nonspecific symptoms. A very small percentage of infected individuals progress to the disease presenting different clinical forms. The acute/subacute form (or juvenile form) predominantly affects children, adolescents and young adults. It is characterized by a rapid clinical evolution and wide dissemination of the fungus to different organs. The chronic form (or adult form), responsible for most cases of PCM, is characterized by a slow clinical evolution, generally affecting adults aged 30 to 60 years and is predominant in males. The main clinical manifestations are due to the impairment of lungs; however, it can cause lesions in the mucosa of the upper aerodigestive tract and in the skin or other organs due to fungal dissemination [1,4,5]. Individuals with PCM may also present additional clinical forms such as the regressive form of the disease, rarely diagnosed, with mild clinical manifestations and clinical regression even without treatment; the mixed form, where there are clinical manifestations of both the acute and chronic forms; the isolated organic form, which affects only one organ, but does not meet the criteria for classification as acute or chronic form; and the residual form, which consists of sequel resulting from PCM, mainly characterized by fibrosis and emphysema [6]

Pattern recognition receptors (PRRs) such as dectin-1, Toll like receptor (TLR)-2 and TLR4 interact with pathogen-associated molecular patterns (PAMPs) of *P. brasiliensis*, which induce innate immune activation. Further, acquired immune responses result in a granulomatous inflammatory process that is responsible for the containment or elimination of the fungus as well as the tissue damage observed in PCM [7]. The human immune response with subsequent cytokine profile is an important factor for the clinical outcome of the disease. In fact, patients with the acute/juvenile form of the disease are presented with a T helper type 2 (Th2) and Th9 profile, with low production of interferon-gamma (IFNγ), high levels of interleukin (IL)-4, IL-5, IL-10 and IL-9, and they do not control the infection [8,9,10]. In the chronic/adult form of the disease, a Th17/Th22 response is predominantly detected in peripheral blood mononuclear cells (PBMCs), with important contribution from Th1 cells, presenting production of tumor necrosis factor (TNF), IL-17, IL-22 and IFNγ [10]. IL-17 has been detected in the inflammatory process in skin and mucosal lesions of patients with PCM [11]. The role of Th17-associated cytokines in *P. brasiliensis* infection has been demonstrated in mice with IL-17 receptor A or IL-6 and IL-23 deficiencies. In these animals, there is impairment in the well-organized granuloma formation and great susceptibility to *P. brasiliensis* infection, suggesting a protective role of Th17 profile [12]. In fact, IL-1β and IL-6 contribute to the development of the Th17 cells, being important for the control of P. brasiliensis infection [12,13,14]. In asymptomatic individuals, there is usually a Th1 response, with production of TNF and IFNγ contributing to the containment of the fungal dissemination [8,10]. However, compared to these individuals that do not develop PCM, patients with the chronic form present higher expression of TNF, IL-17, IL-22 and IFNγ, i.e., these cytokines are associated with severe disease [10]. This difference is probably due to the production of cytokines from the Th17 cells. Although these cells contribute to partial resistance to infection, Th17 cytokines can lead to an intense inflammatory response and, consequently, tissue damage and disease complication [13].

In PBMCs, human monocytes react to pathophysiological changes in human body; thus, these cells are suitable samples when searching for biomarkers as cytokines and genes/molecules involved in their genetic control [15]. Some studies have shown that the cytokines IFNγ and TNF are important to control the growth of *P. brasiliensis* in human monocytes and macrophages [16,17,18]. In turn, human monocytes from healthy individuals exposed to viable yeasts of *P. brasiliensis* produced IL-1β, IL-6, IL-10 and TNF [19]. Sonicated *P. brasiliensis* also induces IL-32 production in PBMCs from healthy individuals, a cytokine associated with the monocyte capacity to control the fungus [20]. Moreover, PBMCs from individuals cured of PCM, stimulated ex vivo with the *P. brasiliensis* gp43 protein, produced increased levels of IL-2, IFNγ and IL-10; however, PBMCs from patients with active PCM (chronic or acute form) produced low levels of IL-2 and IFNγ with substantial amounts of IL-10 in response to stimulation with gp43 [21]. From these studies, Th1 immune response appears to control the *P. brasiliensis* infection while mixed Th responses can contribute for partial resistance and chronic inflammatory process.

Genetic polymorphisms such as single nucleotide polymorphisms (SNPs) can be responsible for variability in the immune responses to infections and, consequently, can be associated with the increase of disease severity or resistance as well as large spectrum of clinical manifestations [22]. Studies have sought to elucidate whether the presence of SNPs can explain the variability in the immune response of individuals with PCM. In this sense, it was shown that the presence of the A allele (genotypes A/A and G/A) in the promoter region of the *TNF* gene (−308 G/A) and *IL10* gene (−1082 G/A) as well as the presence of the C allele (−590 C/T) in the *IL4* gene promoter were related to increased production of these cytokines by PBMCs from patients with PCM [23,24]. In addition, a polymorphism at intron-3 microsatellite region of *IL4* gene seems to be involved in IL-4 production and the different genotypes were associated with risk or resistance to PCM [25]. Another study evaluated SNPs, rs4804803 and rs7975232, in the DC-SIGN receptor gene (*CD209*), which is a C-type lectin receptor, and in the vitamin D receptor gene (*VDR*), respectively. The C allele of rs7975232 and the GG genotype of rs4804803 were associated with a high susceptibility to oral PCM [26]. In addition, Sato et al. [27] observed that the A allele of the SNP rs1946518 (−607) in the *IL18* gene is related to high disease severity in individuals with the acute or chronic multifocal form.

Genetics is one of the host factors that contribute to the interindividual variability during immune responses to infectious microorganisms [28]. The presence of genetic variations regulating the production of innate- and adaptive-derived cytokines induced by *P. brasiliensis* has been poorly investigated. Thus, the aim of the present study was to perform a genome-wide association study (GWAS) to investigate the influence of host genetic polymorphisms in the production of cytokines induced by *P. brasiliensis* in PBMCs from healthy individuals. We describe the potential causal genes controlling IL-1β and IL-22 induced by *P. brasiliensis*.

## 2. Material and Methods

### 2.1. Study Cohort and Ethical Approval

The study was performed in a cohort of healthy individuals of Western ancestry, which are part of the Human Functional Genomics Project (www.humanfunctionalgenomics.org (accessed on 15 June 2020 )) [29]. The age of the individuals goes from 23 to 73 years old, in which 77% are males and 23% are females. 

All volunteers provided written informed consent prior to blood collection and the study was approved by the ethical review board of Radboud University Nijmegen (no. 42561.091.12). In addition, healthy volunteers from Blood Bank of Instituto Goiano de Hematologia e Oncologia (Brazil) participated (only in CLEC7a functional validation experiments) after signing the written informed consent as approved by the Ethical Committee of Hospital de Doenças Tropicais Dr. Anuar Auad-HDT/HAA, CAAE: 81316417.1.3001.0034, Brazil. 

### 2.2. Microorganisms and Culture Conditions

Yeast cells of the *P. brasiliensis* Pb18 strain were grown in complete semi-solid Fava-Netto medium, composed by 1% peptone (Kasvi, São José dos Pinhais, PR, Brazil), 0.5% (*w/w*) yeast extract (BD, Franklin Lakes, NJ, USA), 0.3% (*w/v*) proteose peptone (Sigma-Aldrich, St. Louis, MO, USA), 0.5% (*w/v*) meat extract (BD), 0.5% (*w/v*) NaCl (Sigma-Aldrich), 4% (*w/w*) glucose (Sigma-Aldrich), and 1.2% (*w/v*) agar (BD). The pH was adjusted to 7.2 and the medium was supplemented with 5% (*v/v*) fetal bovine serum (FBS, Cripion, Andradina, SP, Brazil) and 1 μg/mL gentamicin (Sigma-Aldrich). The culture was maintained at 37 °C as described by Bastos et al. [30] and subcultivations were made every seven days.

To obtain heat-killed *P. brasiliensis* for stimulation of PBMCs, the colonies were expanded in an infused medium of the brain and liquid heart (BD) supplemented with 4% (*w/w*) glucose, 5% (*v/v*) FBS, and 1 mg/mL gentamicin (Complete BHI medium). The expansion was maintained in a shaker with 150 rpm, at 37 °C, for three days. After incubation, the yeasts were centrifuged (1400× *g*, 5 min), washed three times with PBS (phosphate-buffered saline solution), and the supernatant was discarded. The quantification was performed on a hematocytometer, and the cell viability was assessed by vital staining with 0.1% (*w/v*) trypan blue in PBS. The yeasts were autoclaved for 40 min at 120 °C and heat-killed yeast cells (4 × 10^7^/mL) were stored at −20 °C. 

The conidia forms of *Aspergillus fumigatus*, strain Af293 (ATCC 46645) and *Rhizopus oryzae*, strain Af1163, were prepared and heat inactivated as described previously [31]. A suspension of 1 × 10^6^ conidia/mL was used in the experiments. 

### 2.3. Peripheral Blood Mononuclear Cells and Cytokine Measurement

To obtain PBMCs, peripheral blood from healthy individuals was collected in tubes with EDTA. Whole blood was centrifuged (200× *g*, 15 min, 4 °C), plasma was discarded, and blood was diluted 1:2 in PBS containing 0.01 mM EDTA (PBS-EDTA) and centrifuged (1400× *g*, 15 min, 4 °C) on Percoll gradient (1.077 density; 61.5%) (Percoll, Sigma-Aldrich). Alternatively, Ficoll (Ficoll-Paque Plus; GE healthcare, Zeist, The Netherlands) gradient centrifugation was used to isolate PBMCs. The mononuclear cell layer was harvested, and the cells were washed twice in PBS-EDTA (600× *g* in the first and 200× *g* in the second centrifugation, 10 min, 4 °C). Then, they were resuspended in RPMI 1640 medium supplemented with 100 U/mL of penicillin (Sigma-Aldrich) and 100 μg/mL of streptomycin (Sigma-Aldrich), 2 mM L-glutamine (Sigma-Aldrich), and 11 mM of sodium bicarbonate (Sigma-Aldrich). The cells were quantified on a hematocytometer and cell viability (≥90%) was assessed using 0.1% (*w/v*) trypan blue. The cell density was adjusted to 5 × 10^6^/mL and 5 × 10^5^ cells in 100 μL/well were seeded in round bottom 96-well plate (Corning, New York, NY, USA). The PBMCs were incubated with heat-killed *P. brasiliensis* (2.5 PBMCs:1 yeast), *A. fumigatus* or *R. oryzae* (2.5 PBMCs:1 conidia) for 24 h or seven days. For seven-day cultures, the medium was supplemented with 10% (*v/v*) autologous serum. After incubation period, the supernatants were collected for cytokine measurement (TNF, IL-6, IL-1β, IL-17, IL-22, IFNγ). Cytokines were measured by using ELISA commercial kits according to manufacturer’s instructions (R&D Systems, Minneapolis, MN, USA). 

### 2.4. Genotyping, Quality Control and Cytokine QTL Mapping

DNA samples of 200 individuals were genotyped using a commercially available SNP chip, Illumina HumanOmniExpressExome-8 v1.0. Genotype calling was performed using Opticall 0.7.0 using default settings [32]. The imputation and quality control were performed as in the previous study [28]. In brief, samples with a call rate ≤ 0.99 were excluded, as were variants with Hardy–Weinberg equilibrium (HWE) ≤ 0.0001, and minor allele frequency (MAF) ≤ 0.1. Strands of variants were aligned and identified against the 1000 Genome reference panel using Genotype Harmonizer. In total, there were 158 samples (Western ancestry) with both DNA and cytokine measurements. Cytokine levels were log-transformed and the R/MatrixEQTL package was used to conduct the genome-wide association. We used a linear regression model with age and gender as covariates. We considered a *p*-value < 5 × 10^−8^ as the genome-wide significant threshold. In addition, a *p*-value of <5 × 10^−6^ was considered as the suggestive threshold for downstream analysis. To investigate the genomic context around genome-wide significant associations, summary statistics for each phenotype were uploaded to the LocusZoom [33] server to visualize regional QTL mapping scan results, using hg19 and 1000 Genomes Nov 2014 EUR as reference genomes. Enrichment analysis of biological pathways was performed using Gene Set Enrichment Analysis (GSEA) [34,35].

### 2.5. Functional Validation of Potential Causal Genes (CLEC7A, PROM1)

To validate potential causal genes related to cytokine variation in response to *P. brasiliensis*, we selected the SNPs that show a *p*-value of <5 × 10^−6^. Among the identified SNPs, rs11053595-*CLEC7A* (encodes Dectin-1) associated with the production of IL-1β, and rs62290169-*PROM1* (encodes Prominin 1 or CD133) associated with the production of IL-22, were selected for further functional validation. 

For blocking the Dectin-1 and Dectin-2 receptors, PBMCs (2.5 × 10^5^ cells/well in 100 μL) were incubated 1 h with 10 μg/mL of anti-Dectin-1 (clone GE2; Catalog MA5-16692; Invitrogen, Thermo Fisher Scientific and clone 259931; Catalog MAB1859; R&D Systems) and anti-dectin-2 (clone #545925; Catalog MAB3114; R&D Systems) neutralizing antibodies or with isotype control antibodies (10 μg/mL) (Catalog I5381; Sigma-Aldrich; IgG2b clone #20116; Catalog MAB004; R&D Systems; IgG1 clone #11711; Catalog MAB002; R&D Systems). For inhibition of downstream signalling pathways molecules, the cells were treated with 1 µM of Raf-1 (GW5074, Sigma-Aldrich) and 50 nM of Syk inhibitor (574711, EMD-Millipore, Burlington, MA, USA). PBMCs were stimulated with heat-killed yeast cells of *P. brasiliensis* (2.5 PBMCs: 1 yeast cell), particulated Pb18 antigens (20 μg/mL; prepared as in Guimarães de Matos et al. [20]), LPS (100 ng/mL, *E. coli* 011:B4, Sigma-Aldrich) or Zymosan (50 μg/mL; Sigma-Aldrich). The cell culture was incubated for 24 h at 37 °C. After incubation, the supernatant was collected to measure IL-1β by immunoenzymatic assay (ELISA), using a commercial kit according to the instructions of manufacturer (BD or R&D Systems).

Flow cytometry was used to determine the CD38 surface expression in PBMCs. Cells from volunteers carrying the different genotypes of the rs62290169 SNP (*PROM1* gene) were used. The cells (2.5 × 10^5^ cells per well in 100 μL) were incubated with heat-killed yeast cells of *P. brasiliensis* (MOI 2.5 cells:1 fungus) for 24 h or seven days. After the incubation period, cells with or without stimulus were labeled with the following antibodies: anti-HLA-DR (clone G46-6, BUV661, BD Biosciences, Franklin Lakes, NJ, USA), anti-CD56 (clone NCAM16.2, BUV737, BD Biosciences), anti-CD3 (clone UCHT1, PerCP, Biolegend, San Diego, CA, USA), anti-CD4 (clone RPA-TA, AF594, Biolegend), anti-CD8 (clone B9.11, PacBlue, Beckman Coulter, Indianapolis, IN, USA), anti-CD45 (clone J3.3, KrO, Beckman Coulter), anti-CXCR5 (clone J252D4, BV605, Beckman Coulter), anti-CD38 (clone HB-7, BV786, Biolegend), anti-CXCR3 (clone G025H7, AF488, Beckman Coulter), anti-CCR4 (clone L291H4, PC5, Beckman Coulter), anti-CCR6 (clone B-R35, PC7, Beckman Coulter) and ViaKrome as viability dye (IR885/40, C36628, Beckman Coulter). Afterwards, the cells were washed and resuspended in FACS buffer (PBS with 2% bovine serum albumin) and a minimum of 9000 and maximum of 16,000 events were acquired based on CD45 expression. The samples were assessed in a CytoFLEX flow cytometer (Beckman Coulter) and the results were analyzed using Kaluza analysis software (Beckman Coulter, IN, USA).

### 2.6. Statistical Analysis

Statistical analyses were performed in R (v3.6.0). The cytokine levels obtained for GWAS were transformed using inverse-normal transformation and the p-value was assessed by Shapiro test. Unsupervised hierarchical clustering was performed using Spearman’s correlation as the measure of similarity. R-package Matrix-eQTL [36] was used for cytokine QTL mapping, where the linear model was applied with age and gender information included as covariates. Manhattan plot and Q-Q plot for each association analysis were generated by package qqman [10.21105/joss.00731] using default parameter settings. In the validation, the results for cytokine IL-1β (pg/mL) are presented as medians with interquartile ranges, analyzed by One-way ANOVA Kruskal–Wallis, followed by post hoc Dunn’s test, Wilcoxon’s paired test, or Mann–Whitney. GraphPad Prism 6.0 Software Inc. (San Diego, CA, USA) was used for the analysis. The level of significance was set at *p* < 0.05.

## 3. Results

### 3.1. The Profile of P. brasiliensis-Induced IL-1β, IL17 and IL-22 Is Similar to Aspergillus fumigatus and Rhizopus oryzae

To determine whether *P. brasiliensis* induced a pathogen specific cytokine profile, PBMCs from 158 volunteers of the 200FG cohort were stimulated for 24 h or seven days, and we assessed the correlation between the levels of cytokines induced by *P. brasiliensis* or *A. fumigatus* and *R. oryzae* as the measure of similarity (Figure 1A). Overall, among the cytokines related to innate immunity (TNF, IL-6 and IL-1β) produced after 24 h of exposure to the three fungi, only the production profile of IL-1β induced by *P. brasiliensis* showed a strong resemblance to *R. oryzae*-induced IL-1β production (Figure 1B). For this cytokine, *R. oryzae* and *A. fumigatus* also presented similar profiles of responses (Figure 1A). It was not detected similarities in the TNF profile induced by the three fungi, but for IL-6, a significant correlation was observed among the three fungi (Figure 1B). For the T-helper cytokines produced after seven days of culture, strong, moderate and weak correlations were detected for IL-17, IL-22 and IFNγ, respectively, among the three fungi (Figure 1A). The production of cytokines induced by each specific fungus, IL-1β and IL-6, were strongly correlated independent of the fungus genus. IL-1β and TNF were strongly correlated when PBMCs were stimulated with *R. oryzae*, but for *P. brasiliensis* this correlation was weak and there was no correlation between these cytokines when *A. fumigatus* was the stimulus. A moderate correlation between IFNγ and IL-22 levels was detected when *P. brasiliensis* or *R. oryzae* stimulated PBMCs, whereas *A. fumigatus* stimulation led to a weak association between these cytokines (Figure 1B). The IL-17 levels were not associated to other cytokines independent of the fungus genus. In addition, no correlations were detected between levels of monocyte- and Th-produced cytokines upon stimulation with any fungal type.

Next, we assessed how the cytokines levels were distributed across the individuals of our cohort upon *P. brasiliensis* exposure by displaying the log-transformed values. We observed a similar distribution of individuals with high and low IL-1β and IL-6 response to *P. brasiliensis*, whereas the majority of the individuals produced low levels of TNF (Figure 1C). Moreover, a substantial number of low responders were seen for IL-17 in comparison to IL-22 and IFNγ (Figure 1D). The inter-individual variability for each cytokine produced after fungal stimulations is shown in Appendix A.

### 3.2. Genome-Wide Mapping Identifies Genetic Variations Associated with Innate- and Adaptive-Derived Cytokines Induced by P. brasiliensis

In order to study the genetic influence in *P. brasiliensis*-induced cytokine production, we performed cytokine production quantitative trait loci (cQTLs) analysis as described in Li et al. [28]. After multiple testing corrections in which the threshold for genome-wide significance was set at <5 × 10^−8^, no genome-wide significant cQTL was identified. To increase our sensitivity for cQTL influencing *P. brasiliensis*-induced cytokine production, we analyzed cQTLs with *p* values < 5 × 10^−6^ for IL-6, IL-1β, INFγ and IL-22 (Table 1). cQTL analysis were not performed for TNF and IL-17 as the levels of these cytokines were very low for most of the individuals evaluated (cytokine production below detection limit). 

We identified an SNP (rs11053595) in the 3′ UTR region of the *CLEC7A* gene (at chromosome 12), which encodes the Dectin-1 protein receptor, associated with IL-1β production (*p* = 5.82 × 10^−7^) upon *P. brasiliensis* exposure (Figure 2A). There was only one homozygous individual for A allele of rs11053595 SNP in which lower amounts of IL-1β were observed after stimulation with heat-killed *P. brasiliensis* yeasts compared to homozygous individuals for C allele (Figure 2B). 

We also identified an SNP (rs62290169) in an intronic region of the *PROM1* gene (at chromosome 4), which encodes the CD133 transmembrane protein. This protein has been used as a marker to isolate cancer stem cells in several tumors. Although it is highly expressed in embryonic stem cells, its function in these cells remains unknown [37,38]. This cQTL was associated with the production of IL-22 (*p* = 2.09 × 10^−7^) upon *P. brasiliensis* exposure (Figure 2C). The homozygous individuals for G allele of rs62290169 SNP produced more amounts of IL-22 after stimulation with heat-killed *P. brasiliensis* yeasts compared to homozygous individual for C allele (Figure 2D). Therefore, the results indicate that the *CLEC7A* and *PROM1* genes are causal genes affected by rs11053595 and rs62290169 SNPs, respectively, whose allelic variations result in disturbances of the cytokine production influencing the innate immune response towards infection caused by *P. brasiliensis*. 

Further, we tested whether the identified cQTLs are associated with biological pathways. Interestingly, among the various pathways enriched for the SNPs that affected IL-1β production, we identified the association of IL-1β with NLRP3 inflammasome and glutathione metabolism (Figure 2E). Moreover, the biological pathways associated with IL-22 QTLs were related to metabolism of vitamins, pantothenate metabolism and CTCF, a protein involved in the regulation of chromatin architecture (Figure 2F). 

### 3.3. IL-1β Production Induced by Heat-Killed Yeasts Cells or Particulated P. brasiliensis Antigens Is Dependent on Dectin-1 Receptor

We have identified an SNP (rs11053595) located in the *CLEC7A* gene (chromosome 12) that influenced the IL-1β production upon *P. brasiliensis* exposure (Figure 3A). eQTL analysis demonstrated the different genotypes of rs11053595 influencing *CLEC7A* expression in whole blood (data obtained from GTEX Portal, www.gtexportal.org (accessed on 24 march 2023)) (Figure 3B). As this gene encodes the Dectin-1 receptor, we performed biological assays to confirm its influence on cytokine production. In fact, blocking of Dectin-1, but not Dectin-2, with neutralizing antibodies significantly decreased the production of IL-1β induced by heat-killed yeasts or particulated *P. brasiliensis* antigens (Figure 3C,D, Appendix A). In Figure 3C, it was shown that, after treatment with anti-Dectin-1, the IL-1β production induced by heat-killed yeasts or particulated antigens, as well as by zymosan, was 60% to 70% inhibited in comparison with controls (IgG control) in most individuals (Figure 3C left). The absolute values for IL-β concentration are also shown (Figure 3C right). In addition, inhibition of Raf-1, but not Syk (*p* value = 0.06; Wilcoxon test), which are both well-known molecules downstream of the dectin-1 pathway [39,40], decreased the production of IL-1β induced by heat-killed *P. brasiliensis* (Figure 3E). Thus, our data show that the rs11053595 SNP affects the function of the Dectin-1 receptor, which in turn is crucial for the induciton of IL-1β during the innate response to *P. brasiliensis*.

### 3.4. The PROM1 rs62290169 SNP Regulates the Surface Expression of CD38 in T-Helper Lymphocytes

We identified an SNP in the intronic region of *PROM1* related with IL-22 production (Figure 4A). However, the *PROM1* gene is a poorly expressed gene in whole blood in comparison to other tissues in the body (Figure 4B). The *PROM1* gene is located in close proximity with the gene that encodes the CD38 molecule, which is a marker of T cell activation [41]. Furthermore, three-dimensional chromatin structures, known as topologically associating domains (TADs), show that chromatin in the eukaryotic nucleus is divided into regions enriched in chromosomal contacts [42]. Publicly available chromosome conformation capture (Hi-C) data obtained from K562 cells (hematopoeietic cell line) revealed the presence of a region containing the CTCF architectural protein (Figure 4C), which is capable of promoting the interaction of DNA by forming loops that brings proximal regions in close proximity [43]. Based on this information, we hypothesized that *PROM1* makes contact with CD38 through CTCF-dependent loop formation. In order to validate this conection, we stimulated PBMCs from health donors with *P. brasiliensis* for 24 h or seven days and evaluated the expression of CD38 by flow cytometry (Appendix A). All gate strategies for flow cytometry evaluation are described in Appendix A. A significant increase in the frequency of CD38^+^ cells was observed after seven days of stimulation with heat-killed *P. brasiliensis* in comparison to non-stimulated cells (Appendix A). Additionally, our results have shown that individuals carrying the GG genotype of the rs62290169 SNP in *PROM1* gene produced higher amounts of IL-22 than CC genotype (Figure 2D). In order to study the relationship between *PROM1* and CD38, healthy individuals carrying the different genotypes of the rs62290169 SNP from the 200FG cohort were evaluated for CD38 expression in T-helper lymphocyte subsets upon *P. brasiliensis* exposure. The results showed that individuals with GG genotype presented higher frequency of CD38^+^ Th1 cells when PBMCs were cultured with *P. brasiliensis* in comparison to non-stimulated cultures (Figure 4D). We did not detect significant alterations in the frequency of CD38^+^ Th2 (Appendix A) or Th17 cells after exposure to the fungus (Figure 4D). Of note, although we only had one individual carrying the CC genotype, we observed that the percentage of Th1, Th2 and Th17 expressing CD38 was lower than GC and GG genotypes. After evaluating CD38 mean fluorescence intensity (MFI) in Th1, Th2 and Th17 cells, the results have shown that as for percentages *P. brasiliensis* exposure led to an increased CD38 MFI in comparison to non-stimulated cells (Appendix A). Moreover, in the individual carrying the CC genotype, the decrease in CD38 MFI was observed only in Th1 cells. Together, our results indicate that CD38 expression in T lymphocytes is modulated by *P. brasiliensis* in PBMCs, a process that appears to be modulated by the contact of *PROM1* and nearby regulatory elements with *CD38*.

## 4. Discussion

The present study aimed to identify host genetic polymorphisms that are involved in the cytokine response of human PBMCs induced by *P. brasiliensis* using GWAS. First, we evaluated the cytokine profile induced by heat-killed Pb18 yeasts in PBMCs from healthy individuals after 24 h and seven days of culture, respectively. After, we performed a correlation test between cytokine levels induced by *P. brasiliensis* and the fungi *A. fumigatus* and *R. oryzae*, which are among the most relevant agents of the fungal human diseases, aspergillosis and mucormycosis, respectively [44,45,46]. Despite these fungi causing distinct diseases, there was a similar profile of IL-1β production induced by *P. brasiliensis* and by *R.oryzae* after 24 h of culture. A similar profile was also detected for IL-17 and IL-22 induced by *P. brasiliensis* and the other fungi in the seven-day cultures, thus indicating that IL-1β, IL-17 and IL-22 are similarly induced by the three fungi evaluated in the present study. On the other hand, the absence of similarity between profiles of TNF and weak correlations for IL-6 and INF-γ production induced by the three fungi suggests that different mechanisms are associated in the induction of these cytokines. Considering the cytokines induced by a specific fungus, for *P. brasiliensis* it is noticeable that there is a strong positive correlation between IL-1β and IL-6 production. The same pattern was observed for *A. fumigatus* and *R. oryzae,* but to a lower extent. Nevertheless, IL-1β and TNF were also strongly associated when *R. oryzae* stimulated PBMCs, whereas in *P. brasiliensis*-stimulated cultures, these cytokines were weakly associated. In opposite, *A. fumigatus* induced IL-1β and TNF, but no correlation was detected between the two cytokines, suggesting that distinct monocyte-derived cytokines can be associated depending on the fungus genus. The production of IFNγ and IL-22 were positively associated after stimulation of the PBMCs with each of the fungus, although for *A. fumigatus,* the correlation was weak. Altogether, these data suggest that PAMPs of the three fungi trigger the production of cytokines involved in Th17/Th22 profiles (IL-1β, IL-17, IL-22), which, together with host genetic factors, can explain the inter-individual variability in the induction of immune responses. 

In this study, by evaluating the variability of cytokine levels in response to heat-killed *P. brasiliensis*-exposed PBMCs, we observed the presence of good and poor responders, ranging from individuals with ability to produce very high or low levels of IL-1β and IL-6 as well as IFNγ and IL-22. The TNF and IL-17 production profiles, on the other hand, were composed by individuals producing very low levels of these cytokines in response to *P. brasiliensis*, therefore impairing the possibility of studying the genetic contribution associated to the inter-individual variation for these cytokines. As genetic polymorphisms can contribute for host cytokine variability [28,47], GWAS analyses were done and, in agreement with previous study [28], we initially set the genome-wide significance threshold at *p*-value < 5 × 10^−8^; however, we did not identify any significant variation, which could be due the fact that we used a small cohort composed by healthy volunteers of Western European ethnicity. Next, we used a suggestion threshold, *p*-value < 5 × 10^−6^, to determine the cQTLs associated with the production of IL-6, IL-1β, IFN-γ and IL-22. We identified several SNPs associated with the production of these cytokines induced by heat-killed *P. brasiliensis*. Among the several cQTL identified, we decided to focus on SNPs associated with IL-1β and IL-22 production, components of Th17/Th22 profiles, which have been associated with resistance to *P. brasiliensis* infection [10,12,48,49,50].

The SNP (rs11053595) is located in the *CLEC7A* gene (3’UTR region), which encodes the Dectin-1 receptor and was significantly associated with IL-1β production. The AA genotype of rs11053595 SNP was related to low amounts of IL-1β after stimulation with heat-killed *P. brasiliensis* yeasts compared to CC genotype. The SNP can influence resistance or susceptibility to PCM since Dectin-1 is known as a receptor that is crucial for the recognition of β-glucan present in the *P. brasiliensis*-cell wall [49,51,52]. We showed that blocking Dectin-1, but not Dectin-2, significantly decreased IL-1β production induced by heat-killed yeast or *P. brasiliensis* particulate antigens. It is known that Dectin-2 triggering by *P. brasiliensis* induces the production of IL-1β in human plasmacytoid dendritic cells (pDCs) [53], which was not observed in PBMCs in the present study. In our experiments, there was a partial inhibition of IL-1β production induced by heat-killed yeast or particulate antigen (60% to 70%) after Dectin-1 blockade, suggesting the participation of other receptors in *P. brasiliensis* recognition. 

Our results showed that *P. brasiliensis*-induced IL-1β production was dependent on Raf-1 signaling pathway (*p* < 0.05), although Syk appears to participate in less extension (*p* value = 0.06). It is known that Syk and Raf-1 are independent pathways that converge at NF-κB activation level to induce cytokine production after Dectin-1 triggering in human DCs [39]. Moreover, inhibition of Syk completely abolishes the production of cytokine induced through activation of Dectin-1 alone, but when TLR is involved in the activation, Raf-1 mediates the Dectin-1 signaling [39,54]. Thus, our results suggest that Dectin-1 can be acting together TLR to induce IL-1β in PBMCs exposed to heat-killed *P. brasiliensis*. In human monocytes and human and mouse DCs, the evaluation of Dectin-1/TLR2/TLR4 in the immune response against *P. brasiliensis* has been reported, but the cooperation among these receptors to the production of IL-1β is missing [49,55,56]. However, de Castro et al. [49] showed that blocking only TLR2 in human DCs stimulated with *P. brasiliensis* had no effect on IL-1β production, whereas Dectin-1/Syk pathway was crucial for NLRP3 inflammasome-dependent cytokine production (IL-1β, IL-18). These results suggest that *P. brasiliensis* acting on TLR2 and Dectin-1 can induce pro-IL-1β through NFkB pathway, but only Dectin-1/Syk is crucial for inflammasome activation and IL-1β release [49]. Further studies aiming to investigate pro-IL-1β and bioactive IL-1β production can reveal the most relevant pathways to induce higher levels of IL-1β after stimulation with *P. brasiliensis* than with other fungi, as shown here for *R. oryzae* and *A. fumigatus*. To our knowledge, no study has been performed with *R. oryzae* and inflammasome; however, *A. fumigatus* activates two inflammasomes, AIM2 and NLRP3, to induce IL-1β production and fungal control [57,58,59]. Thus, additional experiments should be performed to elucidate possible cooperation between innate receptors for IL-1β production during the immune response against *P. brasiliensis* as well as the mechanisms responsible for different levels of IL-1β induced by *P. brasiliensis* and *A. fumigatus*. 

Our results confirmed Dectin-1 as an important receptor for IL-1β production during the response to *P. brasiliensis*. Other studies have also shown the relevance of Dectin-1 in this fungal infection. Loures et al. [60] showed that Dectin-1 plays an important protective role in the lungs during infection with *P. brasiliensis* in mice. Dectin-1 deficiency impaired murine macrophage fungicidal mechanisms by decreasing nitric oxide (NO) and increasing IL-10. The disease was severe in the absence of Dectin-1, with increased lung pathology and mortality. Deficiency in Dectin-1 also contributed to a decrease in Th1/Th2/Th17 cytokines, decreased Th17 cell differentiation and increased T regulatory cell expansion. Ex vivo, Silva et al. [61] blocked Dectin-1 in human monocytes and neutrophils before infection with *P. brasiliensis* yeasts, which resulted in a decreased production of hydrogen peroxide and increased fungal load in these cells. As IL-1β is produced after Dectin-1 activation, this cytokine can be at least partially responsible for Dectin-1 effects in the fungal resistance. Tavares et al. [14] showed that IL-1β-induced signaling is important for the control of *P. brasiliensis* infection in murine macrophages. This cytokine also plays an important role in the inflammatory process and contributes to the development of a Th17/Th22 responses [13,62], which appears to contribute for *P. brasiliensis* infection resistance [10,12,50]. 

It has been demonstrated by eQTL analysis that the different genotypes of rs11053595 SNP impact the expression of *CLEC7A* in whole blood (data obtained from the GTEX Portal, www.gtexportal.org), despite the SNP being located at the 3’UTR. It is possible that this difference in the expression of *CLEC7A* is related to differences in mRNA stability, in the degradation process, as well as in the translation of this mRNA, thus regulating the levels of dectin-1. Additionally, it is possible that the different genotypes present mRNA isoforms containing different 3’UTRs, which can lead to post-translational modifications as protein conformation [63].

Strengthening our data about the relevance of the SNP in *CLEC7A* and IL-1β production, when we analyzed whether cQTLs were associated with biological pathways, we identified the association of IL-1β cQTL with the NLRP3 inflammasome and glutathione metabolism pathways. Dectin-1 activation is related to the NLRP3 inflammasome activation. After the recognition of β-glucan by Dectin-1/Syk phosphorylation, NLRP3 inflammasome is activated and, in turn, activates caspase-1 that processes pro-IL-1β into bioactive IL-1β [14,49,51]. Dectin-1/caspase 8 pathway also regulates IL-1β production during *P. brasiliensis* infection [64]. This pathway has been reported in *P. brasiliensis* infection as a mechanism that is critical for resistance against the fungus by inducing IL-1β and promoting Th1/Th17 responses [14,48,51,64]. In turn, glutathione metabolism is important in several biological processes such as protection against oxidative stress caused by reactive oxygen species (ROS), since glutathione in its reduced form interacts with reactive metabolites and helps in their removal [65,66]. In fungal infection, Seegren et al. [67] showed that, during phagocytosis of *Candida albicans* and zymosan by murine macrophages, there was stimulation of ROS production and this was crucial for fungal control. Furthermore, the study of Santos et al. [68] shows that the *P. brasiliensis* Pb18 strain is capable of inducing ROS in murine polymorphonuclear cells and Tavares et al. [14] showed that this fungus induces ROS/inflammasome in murine macrophages. Therefore, we could speculate that heat-killed *P. brasilienis* yeasts can induce ROS production in PBMCs which, consequently, leads to activation of the NLRP3 inflammasome and glutathione metabolism. 

Associated with IL-22 production, we identified an SNP (rs62290169) located in the *PROM1* gene, which encodes the CD133 protein. Despite this protein being highly expressed in embryonic stem cells, its function in these cells remains unknown [37,38]. The identified SNP is located in a non-coding and junction region with the gene encoding the CD38 molecule. In lymphocytes, CD38 is a glycoprotein that acts as an enzyme and a multifunctional receptor, which is important in the processes of activation, proliferation, and secretion of cytokines [69]. CD38 also plays a role in DC migration and survival as well as Th1 differentiation profile [70]. It was shown that, in the cQTL where *PROM1* is located, there is a region that contains the CTCF architectural protein, which is capable of promoting DNA interaction through the formation of loops [43]. As the topologically associated domain (TAD) map showed a connection between CD38 with *PROM1* and it was shown that CD38 can control IL-22 production [71], we hypothesize that *PROM1* makes contact with *CD38* through CTCF-dependent loop formation to control production of IL-22. 

To validate our hypothesis, we evaluated the expression of CD38 on T cells from healthy individuals with the different *PROM1* SNP rs62290169 genotypes after stimulation with heat-killed *P. brasiliensis* yeast. The 7-day exposure to the fungus induced a significant increase in CD38 expression compared to the negative control. Flow cytometry showed that the individual carrying the CC genotype had lower percentages of CD38^+^ Th1, Th2 and Th17 lymphocytes, suggesting that SNP *PROM1* C allele can affect CD38 expression. In addition, CD38 MFI in Th1 cells of the individual carrying the CC genotype was shown to have the same pattern observed for percentages, whereas this was not the case for Th2 and Th17 cells. We had shown that the individual carrying the CC genotype produced lower levels of IL-22 than homozygous GG individuals. Consistent with our results, Gangemi et al. [71] showed that patients with chronic B lymphocytic leukemia who had high CD38 expression presented higher levels of IL-22 than patients with low CD38 expression. Therefore, our data suggest that during *P. brasiliensis* infection *PROM1* variant rs62290169 can down regulate the CD38 expression, which leads to a decrease of the IL-22 production. Concerning Th lymphocyte subsets, however, CD38 was highly expressed on Th1 cells from GG carriers after *P. brasililensis* exposure. As we did not identify the cell source of IL-22 in our experiments, *PROM1* can regulate the production of IL-22 in Th1, Th22, Th17 or even other cells present in PBMCs as potential sources of IL-22 [72,73]; moreover, this gene can be also connected with IFNγ production, via or not CD38 [70], since Th1 cells produce IFNγ and IL-22 [73,74]. These suggestions need to be further evaluated as well as to investigate whether and how the *PROM1* really controls CD38 expression. We were not able to detect *PROM1* mRNA expression in PBMCs, confirming that this gene is too low expressed in whole blood (Figure 4). 

The biological pathways associated with IL-22 cQTLs were related to vitamin metabolism, pantothenate (vitamin B5) metabolism and the CTCF architectural protein. IL-22 is a cytokine produced by Th lymphocyte subsets and innate lymphoid cells, which is essential for host defense in skin and mucosal immune responses as well as tissue repair [73,75]. It is known that adequate intake of vitamins is important for the proper functioning of the immune system [76]. Cha et al. [77] demonstrated that vitamin A deficiency in a murine model impairs the differentiation of T cells towards a Th17 profile, reducing the expression of cytokines as IL-22. Our group has also recently demonstrated that vitamin D can improve the fungicidal activity of PBMCs infected with *P. brasiliensis* [20]. There are only two studies about *P. brasiliensis* infection and IL-22. In mice, reduction of Th22 cells was associated with an increase of fungal burden in the lungs, enhanced lung pathology and mortality, thus suggesting a protective role for Th22 in pulmonary experimental PCM [50]. Evidence that IL-22 is produced after *P. brasiliensis* infection was shown by de Castro et al. [78] in co-cultures of human macrophages and T lymphocytes. This scarce information about IL-22 in PCM, together our results, indicates that evaluation about the role of this cytokine in this fungal disease deserves further study.

One of the limitations of the present study is the fact that we performed genome-wide association analysis in a cohort consisting of a relatively small sample size, which might explain the absence of the identification of genome-wide significant loci. In addition, the use of heat-killed *P. brasiliensis* cannot represent the overall immune response to the live fungus, and only one strain of *P. brasiliensis* (the Pb18, considered of intermediate/high virulence) was used, whereas some studies have shown differences in virulence among strains in mouse infection as a factor that can influence the induction of cytokines [19,52,79,80]. Another limitation in this study is that the evaluation of patients with PCM to check the influence of the SNPs studied here as well as other *CLEC7A* gene SNPs is missing. Some SNPs in *CLEC7A* gene can influence the expression of different dectin-1 isoforms as well as alter the receptor ability to interact with the ligand [81,82,83]. The SNP rs16910526 (Y238X) leads to an early stop codon, impairing the binding of β-glucan to the receptor and the production of cytokines during the response to *C. albicans* [81]. Therefore, the evaluation of *CLEC7A* SNPs in a cohort of patients with PCM is crucial to determine the contribution of Dectin-1 polymorphisms to *P. brasiliensis* infection. In addition, besides nonopsonic phagocytosis mediated by β-glucan/Dectin-1, opsonophagocytosis mediated by complement and pentraxin are relevant in fungal infections [84]. In human neutrophils, intracellular expression of Dectin-1 can be upregulated by *C. albicans* opsonized by mannose binding lectin (MBL); intracellular Dectin-1 binds to β-glucan, which increases ROS [85]. Thus, in addition to the contribution of complement activation to increase opsonophagocytosis of *P. brasiliensis* [86], the effects of *CLEC7A* SNPs in Dectin-1-mediated phagocytosis in concert with other receptors to fungal opsonophagocytosis deserves further evaluation.

Despite the difficulties in addressing all these questions, the findings of the present study are strengthened by the addition of functional data; moreover, the genetic variability among *Paracoccidioides* spp. has been shown not to influence the development of PCM clinical forms [52]. Lastly, the profile of cytokines induced by heat-killed Pb18 was similar to what is detected in PBMCs from patients with adult/chronic PCM, the most common disease form, where mixed Th1/Th17/Th22 responses are reported [10].

## 5. Conclusions

In conclusion, our results showed the influence of the *CLEC7A* and *PROM1* genes on the production of cytokines IL-1β and IL-22, respectively, in response to *P. brasiliensis*. A more detailed study should be carried out to better understand the role of CD38 in PCM and to elucidate the biological pathways related to IL-22 production. The role of IL-22 in PCM is not yet completely established. As *PROM1* is highly expressed in the lungs (Figure 4B) and associated with the production of IL-22 as well as IL-1β induces IL-22 [87], to evaluate the *PROM1* and *CLEC7A* variants described in this study in PCM patients, future work should contribute to understanding mechanisms involved in resistance or susceptibility to the disease, revealing novel targets for treatment of PCM. In addition, other possible causal genes associated with *P. brasiliensis*-induced cytokine production identified in this study can be validated in future studies showing other genetic variants as determinants of susceptibility or resistance to PCM. Thus, our future aims are to validate other putative causal genes identified in this study, to better investigate the relationship between PROM1, CD38 and IL-22 in immune response to *P. brasiliensis,* and to explain how the rs11053595 SNP in the *CLEC7A* gene influences the function of the dectin-1.

## Figures and Tables

**Figure 1 jof-09-00428-f001:**
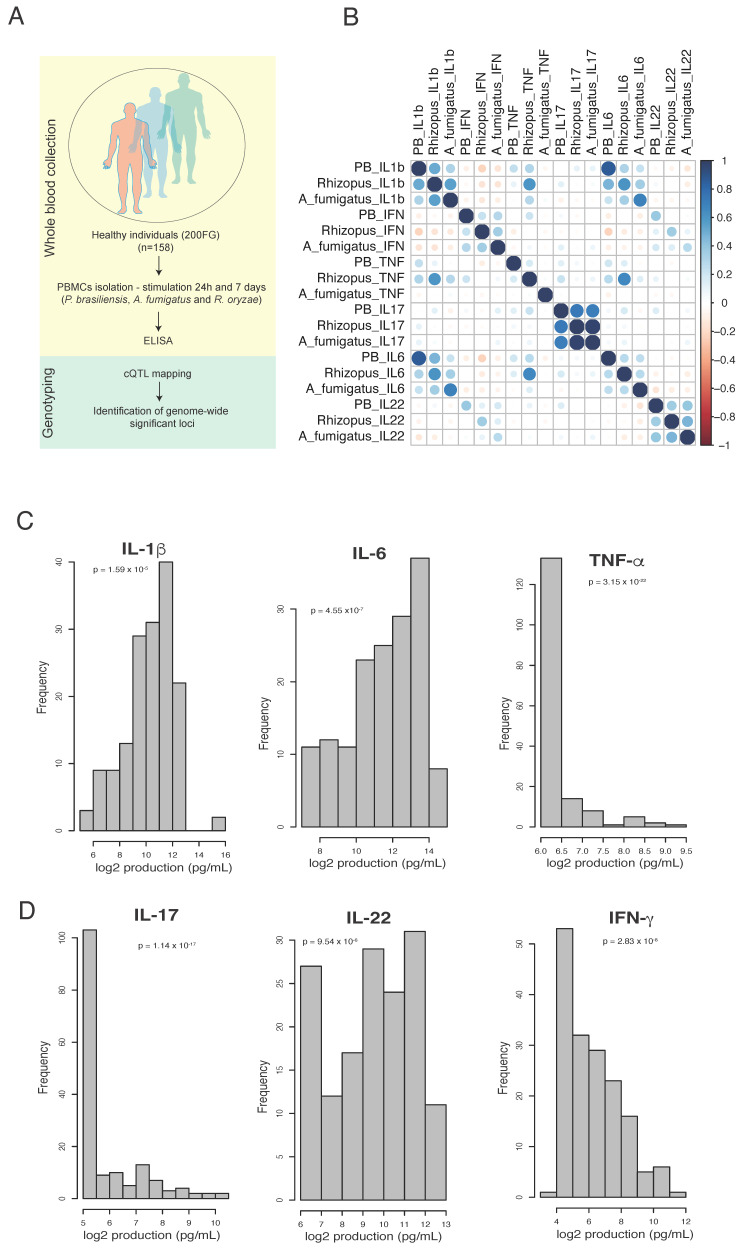
Similarities in innate and acquired cytokine profiles induced by *P. brasiliensis* and other fungi. (**A**) Overview of the study. (**B**) Spearman correlation coefficients between IL-1β, TNF, IL-6 (24 h of incubation), IFNγ, IL-22 and IL-17 (seven days of incubation) levels after PBMC stimulation with heat-killed yeasts of *P. brasiliensis,* conidia of *A. fumigatus* and or *R. oryzae* (2.5 PBMCs:1 fungus; cytokines were measured by ELISA in pg/mL); (**C**,**D**) Distribution of *P. brasiliensis*-induced cytokines after 24 h (**C**) and seven days (**D**), Log2-transformed values were used, and p-value was assessed by Shapiro test.

**Figure 2 jof-09-00428-f002:**
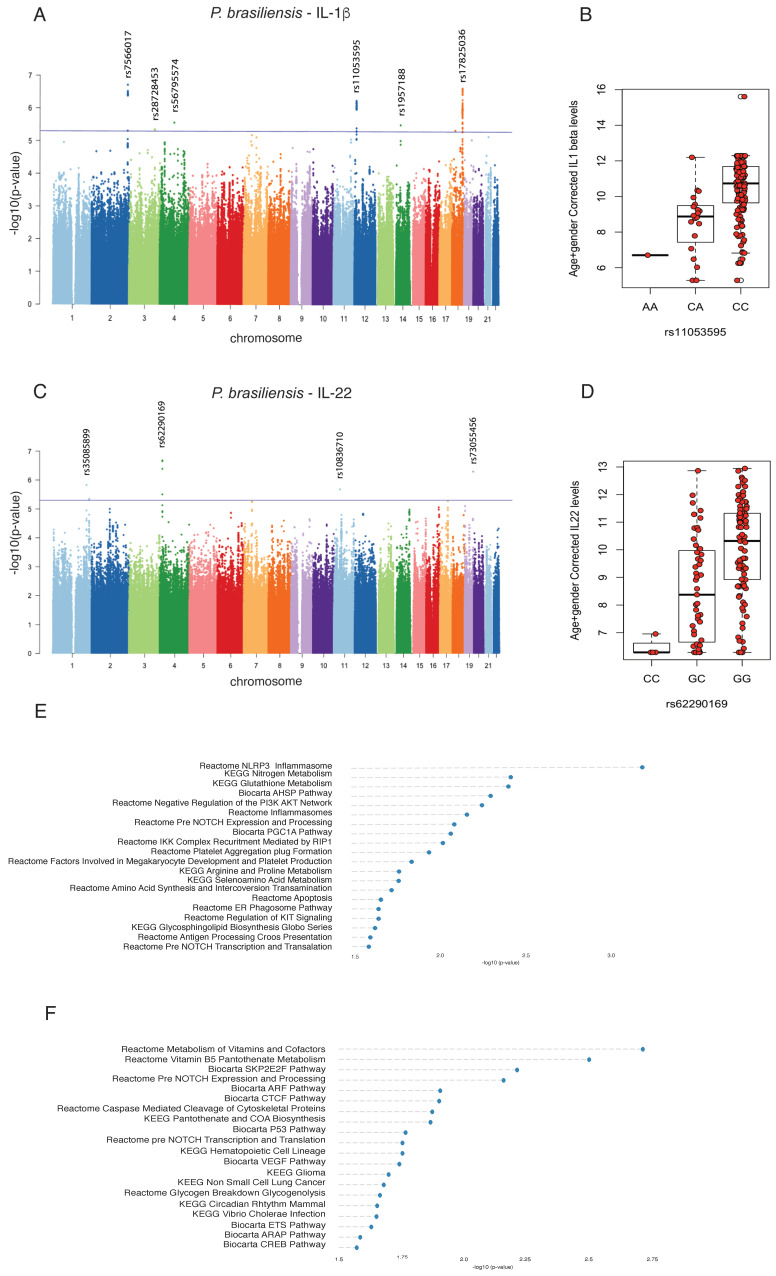
Genome-wide QTL mapping and association genotype with IL-1β and IL-22 levels induced by *Paracoccidioides brasiliensis*, and enriched immune pathways. (**A**,**C**) Manhattan plots showing the cQTL mapping results for *P. brasiliensis*-induced IL-1β and IL22, respectively. The *y*-axis represents −log10 *p*-values and the *x*-axis shows the data of each chromosome. Horizontal blue lines correspond to *p*-value < 5 × 10^−6^ for both IL-1β and IL-22 Manhattan plots. (**B**,**D**) Log2 of age- and gender-corrected cytokine production capacity in the different genotype groups for SNP rs11053595 (*CLEC7A* gene) (*p* value *=* 1.873 × 10^−5^; Kruskal–Wallis test) and rs62290169 (*PROM1* gene) (*p* value *=* 4.429 × 10^−6^; Kruskal–Wallis test). The data is displayed as box-plots with median, quartiles with maximal and minimal values; (**E**,**F**) Enrichment analysis of biological pathways associated to IL-1β and IL-22 QTL.

**Figure 3 jof-09-00428-f003:**
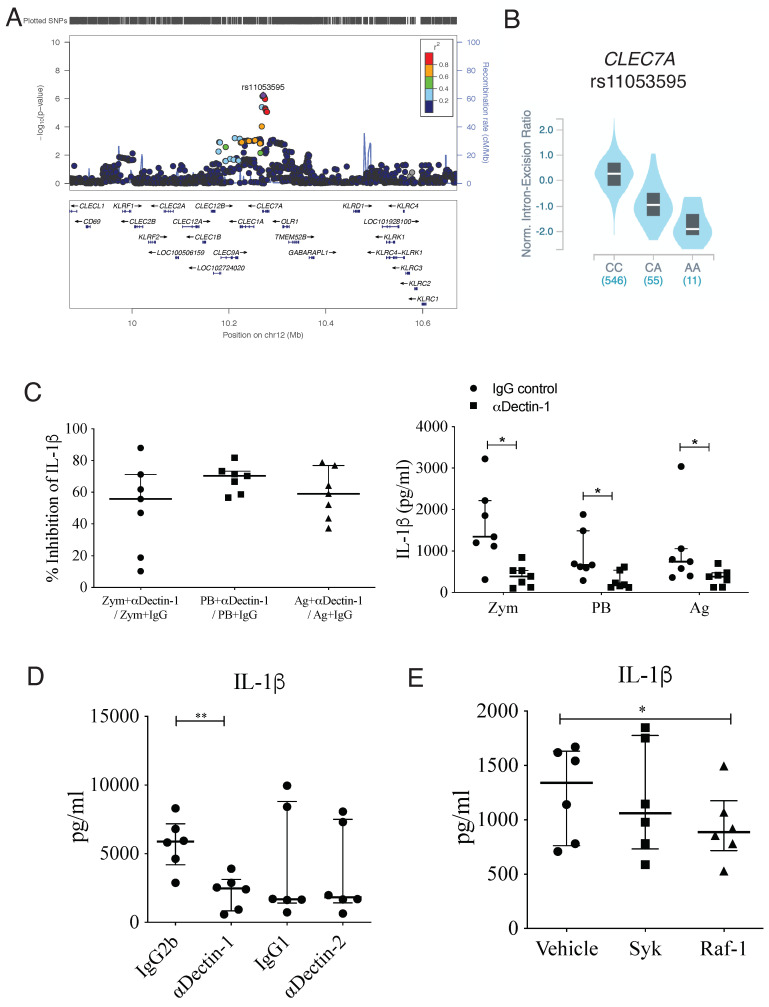
Regional association plot for IL-1β QTL and relevance of Dectin-1 for IL-1β induced by *Paracoccidioides brasililensis*. (**A**) The regional association plot for the IL-1β production-associated SNP (rs11053595, purple diamond), which is located in the 3′ UTR region of the *CLEC7A* gene at chromosome 12. Each dot represents an SNP, and the linkage disequilibrium of neighboring SNPs with the top SNP is color-indicated. The *y*-axis represents −log10 *p* values of SNPs. The *x*-axis shows chromosome 12 positions. (**B**) Whole blood GTEX expression data obtained from individuals carrying different genotypes of the rs11053595 SNP (www.gtexportal.org (accessed on 24 March 2023)). (**C**) Inhibition of IL-1β by neutralizing Dectin-1 antibodies for each stimulus. PBMCs (2.5 × 10^5^ cells/well) obtained from healthy donors were incubated in the absence or presence of neutralizing anti-Dectin-1 antibodies (αDectin-1 at 10 μg/mL) or control IgG antibodies (IgG at 10 μg/mL) for 1 h. After, the cells were stimulated with 50 μg/mL of Zymosan (Zym), 1 × 10^6^ yeasts/mL (2.5 PBMCs:1 yeast) of heat-killed yeasts *P. brasiliensis* (PB), or 20 μg/mL of particulate antigens from *P. brasiliensis* yeasts (Ag) for 24 h. The supernatant was collected to measure the amount of IL-1β by ELISA. (n = 7). On the left, results showed the inhibition of IL-1β production (stimulus + αDectin-1) in relation to the IgG antibody control (stimulus + IgG); on the right are shown the absolute values of IL-1β concentration after Dectin-1 blocking. (**D**,**E**) PBMCs (2.5 × 10^5^ cells/well) obtained from healthy donors were incubated in the absence or presence of neutralizing anti-Dectin antibodies (αDectin-1and αDectin-2 at 10 μg/mL) or isotype control antibodies (IgG2b and IgG1 at 10 μg/mL), 1 µM of Raf-1, 50 nM of Syk inhibitor or vehicle for 1 h. Cells were stimulated with heat-killed yeasts of *P. brasiliensis* (2.5 PBMCs:1 yeast) for 24 h and IL1β production was assessed in the supernatant by ELISA. (n = 6). The data are presented as individual values and medians and interquartile ranges (Wilcoxon paired test, * *p* < 0.05, ** *p* < 0.001).

**Figure 4 jof-09-00428-f004:**
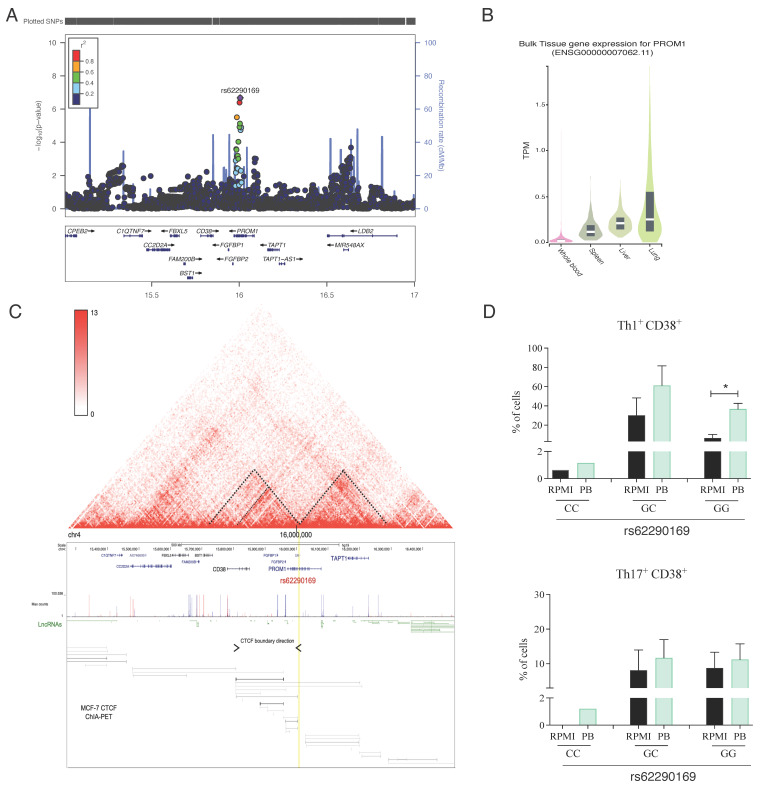
Regional association plot for IL22 QTL and regulation of *Paracoccidioides brasiliensis*-induced CD38 expression by *PROM1* gene. (**A**) The regional association plot for the IL-22 production-associated SNP, rs62290169 (purple diamond), which is located at the intronic region of the *PROM1* gene at chromosome 4. Each dot represents an SNP, and the linkage disequilibrium of neighboring SNPs with the top SNP is color-indicated. The *y*-axis represents –log10 *p* values of SNPs. The *x*-axis shows chromosome 5 positions. (**B**) GTEX expression data obtained from different tissues (www.gtexportal.org accessed on 24 March 2023). (**C**) Topologically associating domains (TADs) of *PROM1* locus. (**D**) CD38 protein expression upon 7 days *P. brasiliensis* exposure assessed by flow cytometry in Th1, and Th17 cells of individuals part of the 200 FG cohort carrying the different genotypes of the rs62290169 SNP (*PROM1* gene) CC (n = 1), GC (n = 4), GG (n = 6) (Wilcoxon paired test, * *p* < 0.05).

**Table 1 jof-09-00428-t001:** Results of cQTL mapping analyses (*p* < 5 × 10^−6^).

SNP	Cytokine	*p* Value	CHROM	POS	REF	ALT	Gene
rs56795574	IL-6	6.90 × 10^−8^	4	91745027	A	G	*CCSER1*
rs10850949	IL-6	1.94 × 10^−7^	12	118442266	G	T	*RFC5*
rs17058173	IL-6	2.20 × 10^−7^	18	73512093	C	T	*LOC339298*
rs35049004	IL-6	9.03 × 10^−7^	16	76913364	G	A	*MIR4719*
rs76262774	IL-6	3.04 × 10^−6^	17	33048609	C	T	*TMEM132E*
rs7566017	IL-1 beta	1.92 × 10^−7^	2	231536848	C	T	*LOC151475*
rs17825036	IL-1 beta	2.39 × 10^−7^	18	65170712	T	G	*DSEL*
rs11053595	IL-1 beta	5.82 × 10^−7^	12	10270822	C	A	*CLEC7A*
rs56795574	IL-1 beta	2.75 × 10^−6^	4	91745027	A	G	*CCSER1*
rs28728453	IL-1 beta	4.42 × 10^−6^	3	163725033	A	G	*MIR1263*
rs1957188	IL-1 beta	9.59 × 10^−6^	14	46509206	T	C	*RPL10L*
rs369304721	IFN gamma	1.34 × 10^−6^	22	51221190	G	A	*RABL2B*
rs111342688	IFN gamma	1.58 × 10^−6^	5	78330613	T	G	*DMGDH*
rs7185240	IFN gamma	2.29 × 10^−6^	16	8272911	G	A	*TMEM114*
rs2245968	IFN gamma	2.43 × 10^−6^	10	7528542	T	G	*TIH5*
rs74439139	IFN gamma	3.23 × 10^−6^	7	113757643	T	G	*FOXP2*
rs62290169	IL-22	2.09 × 10^−7^	4	16005374	G	C	*PROM1*
rs73055456	IL-22	5.10 × 10^−7^	19	54794580	G	C	*LILRA3*
rs35085899	IL-22	1.47 × 10^−6^	1	216889197	T	C	*ESRRG*
rs7252511	IL-22	7.98 × 10^−6^	19	2396243	C	T	*TMPRSS9*
rs10836710	IL-22	2.11 × 10^−6^	11	37256074	T	G	*SNORA31*

Abbreviations: CHROM, chromosome; IL, interleukin; IFN, Interferon SNP, single-nucleotide polymorphisms; POS, genomic position; REF, reference allele; ALT, altered allele.

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
