# Peer review of "Genome-Wide Association Study Reveals CLEC7A and PROM1 as Potential Regulators of Paracoccidioides brasiliensis-Induction of Cytokine Production in Peripheral Blood Mononuclear Cells"

_jof, 2023, doi:10.3390/jof9040428_

Round 1
Reviewer 1 Report
This work deals with assessment a genetic variants associated with PBMCs-derived cytokines produced as reaction to presence of P. b.
158 individuals were enrolled in this experiment.
The work reports that CLEC7 A and PROM1 protein genes are pivotal for the cytokine response with association to PCM progression.
Remarks
1. In introduction section (3rd paragraph), I miss strongly a general statement of the importance of PBMCs as a valid sample where important biomolecular changes happen as a reaction to patho-physiological changes in the body. The searching for important biomarkers is thus feasible.
“Human monocytes react to patho-physiological changes in human body, thus are suitable samples when searching for biomarkers [https://doi.org/10.1002/pmic.202200026]. Some studies have shown that the cytokines IFNγ and TNF are important to control the growth of P. brasiliensis in human monocytes and macrophages [13-15].
2. In the experimental part, provide a figure of the experimental workflow, including the main parts such as sampling from enrolled participants, sample treatment, analysis and evaluation of the acquired data. This would help for the reader for the quick understanding the experimental design used in this work.
3. In Conclusion part, provide the future aims of the authors in this area of investigation.
Reviewer 2 Report
The study aims to detect SNPs showing a strong influence on the cytokine-signature induced by Paracoccidioides brasiliensis as compared to Rhyzopus oryzae and Aspergillus fumigatus. While a strong correlation between the results observed in response to Paracoccidioides and Rhyzopus was observed, this was not the case of Aspergillus. Moreover, TNF expression showed a different pattern from that observed for IL-1β and IL-6. This finding poses a host of mechanistic questions that have not been addressed in the study, since its main purpose was the search of SNPs through quantitative trait loci analysis of several cytokine genes. A major finding of this approach was the identification of the SNP rs11053595 in the 3'-UTR region of the CLEC7A gene, which encodes the Dectin-1 protein receptor and is associated with IL-1β production. A corollary to the study could be that the usage of different sets of transcription factors could explain the distinct findings associated with the induction of cytokine expression. In line with this, while TNF is an archetypal κB-dependent gene, optimal IL1B induction expression requires the concurs of other transcription factors, for instance, HIF-1. After getting the aforementioned result, some mechanistic studies were conducted to obtain relevant information regarding signaling through dectin-1 and how this might influence the course of Paracoccidioides brasiliensis infection.
Specific comments
1. The location of the polymorphism at the 3´-UTR region poses the question of how this influences dectin-1 function since it locates out from the ORF. Expression of CLEC7A mRNA could shed light on the mechanism involved since the SNP does not explain either loss- or gain-of-function, but rather stability of the mRNA or alternative splicing of CLEC7A exons.
2. The paragraph stating that dectin-1 deficiency impaired macrophage fungicidal mechanisms by decreasing nitric oxide (NO) and increasing IL-10 should be fine-tuned since NO production by inducible NOS2 is characteristic of mice macrophages and not of human macrophages.
3. The section devoted to the inhibition of IL-1β production by antibodies is not clear, as regards the results shown in Fig. 3B. The same comment can be applied to the inhibition by zymosan.
4. The inhibition of the effect of Paracoccidioides antigen by anti-dectin-1 Ab should be developed since it indicates either presence of β-glucans in the preparation or activation of dectin-1 by components other than β-glucans.
5. The section devoted to dectin-1 signaling should be reworded since NF-κB signaling is more dependent on TLR2 than on dectin-1 and optimal response involves the collaboration of both receptors.
6. A mention to the role of CLEC7A polymorphisms on dectin-1 signaling should be mentioned, since in addition to loss-of-function associated with the generation of an early stop codon (Y238X), other polymorphisms can influence the expression of dectin-1 isoforms. Moreover, in addition to non-opsonic phagocytosis of fungi, opsonic phagocytosis associated with complement-derived opsonins and pentraxin have been associated with the outcome of fungal infections. Although this has not been explored in the study, a comment on this can be added to the section dedicated to limitations of the study.
7. The allusion to ROS production can be strengthened by the inclusion of references showing strong activation of NADPH oxidases during zymosan and Candida phagocytosis (Seegren et al., Cell Rep. 33: 108411).
8. The enhanced induction of IL-1β produced by Paracoccidioides as compared to Rhyzopus and Aspergillus prompts the study of pro-IL1B mRNA. Should Rhyzopus and Aspergillus induce pro-IL-1β mRNA, then the distinct effect of Paracoccidioides on IL-1β can be explained by its unique ability to activate the inflammasome.
Round 2
Reviewer 1 Report
Authors have reacted to given queries.
Author Response
Academic editor’s comments
1) The revised version of the manuscript still has a few major gaps that need to be addressed:
The authors explain, on page 2, that the acute/juvenile form of PCM is marked by low IFN-g and high levels of IL-4, IL-5, IL-9 and IL-10, whereas the chronic/adult form is marked by TNF, IL-17, IL-22, and IFN-g. For the chronic/adult form, as written, it would imply that excessive TNF, IL-17, IL-22, and IFN-g are associated with severe disease, presumably compared to age/sex-matched controls that are living in the same endemic area but do not develop chronic/adult PCM. Can the authors please clarify if this is correct? If so, please explicitly state that in the intro. If the above cytokines are those that the authors state are related to the different form of PCM, then this would explain why they measured certain cytokines (e.g. TNF, IL-17, IL-22, IFN-g), but they have not explained why they measured others (i.e. IL1b, IL6). This is important to clarify, because IL1b is the premise for the Dectin 1 work.
Answer: We thank the editor for the careful revision of the manuscript, we agree with the suggestions, therefore the missing information is now added in the manuscript as indicated by the gray color:
In the chronic/adult form of the disease, a Th17/Th22 response is predominantly detected in peripheral blood mononuclear cells (PBMCs), with important contribution from Th1 cells, presenting production of tumor necrosis factor (TNF), IL-17, IL-22 and IFNg [10]. IL-17 has been detected in inflammatory process in skin and mucosal lesions of patients with PCM [11]. The role of Th17-associated cytokines in P. brasiliensis infection has been demonstrated in mice with IL-17 receptor A or IL-6 and IL-23 deficiencies. In these animals, there is impairment in the well-organized granuloma formation and great susceptibility to P. brasiliensis infection, suggesting a protective role of Th17 profile [12]. In fact, IL-1b and IL-6 contributes to the development of the Th17 cells, being important for the control of P. brasiliensis infection [12-14]. In asymptomatic individuals, there is usually a Th1 response, with production of TNF and IFNg contributing to the containment of the fungal dissemination [8,10]. However, compared to these individuals that do not develop PCM, patients with the chronic form present higher expression of TNF, IL-17, IL-22 and IFNg, i.e., these cytokines are associated with severe disease [10]. This difference is probably due to the production of cytokines from the Th17 cells. Although these cells contributes to partial resistance to infection, Th17 cytokines can lead to an intense inflammatory response and, consequently, tissue damage and disease complication [13].
2) The identification of a SNP in Dectin-1 requires more work.
The authors identified that the rs11053595 C/A SNP in Dectin-1 correlated with IL1b production. Then, they showed that blocking Dectin-1 with an antibody prevents IL1b production. However, there are a few steps in between these two that need to be filled: The authors need to show how the SNP affects Dectin-1. They absolutely should measure the cell surface expression of Dectin-1, and see if the SNP correlates with surface expression. This may be done on patients' cells or in a cell line transfected with the corresponding construct carrying this 3' UTR SNP. IF a SNP allele does affect surface expression, then the experiment with the blocking antibody makes sense to include. If the SNP allele does not affect surface expression, the blocking antibody experiment is irrelevant to the SNP finding. Indeed, blocking Dectin1 with antibody will affect IL1b and many other cytokines (including TNF), but that is not pertinent to the SNP. In fact, if the SNP does not affect surface expression, the authors should look at Dectin1 function, such as binding to fungal articles.
Answer: We thank the editor for all relevant suggestions on how to further establish the connection between the identified and SNP and dectin-1 function. The points raised by the editor are especially relevant in the context of the future we intend to do with PCM patients. It is important to knowledge the actual challenges in obtaining sufficient numbers of patients bearing the SNP, this process will require time due to the low number of PCM cases in the Brazilian region where this is study has been performed. As a first step, the current study is unique in the sense we’ve performed experiments with P. brasiliensis antigens and made use of functional genomic approach to insights on the contribution of host genetics factors associated to cytokine production. We will certainly consider the execution of the validation experiments required to prove causality. In fact, we finished Conclusions section stating this as:
...and explain how the rs11053595 SNP in the CLEC7A gene influences the function of the dectin-1.
The authors describe PCM and its clinical syndromes, and how genotypes may contribute to these clinical manifestations. Thus, the reader anticipates that there will be SNP-based immunologic findings that correlate with disease. In fact, this is not the case. Of 200 individuals, only 158 were tested because they had both genotypic and cytokine data. Of these 158, it is not clear how many are "healthy individuals of Western ancestry" and how many are "healthy volunteers from Brazil". So, it is unclear how ancestry contributes to the identification (or lack of identification) of SNPs.
Answer: Thanks for pointing out the missing information. We have added the following information (in gray) in Material & Methods:
2.1 Study cohort
All volunteers provided written informed consent prior to blood collection and the study was approved by the ethical review board of Radboud University Nijmegen (no. 42561.091.12). In addition, healthy volunteers from Blood Bank of Instituto Goiano de Hematologia e Oncologia (Brazil) participated (only in CLEC7a functional validation experiments) after signing the written informed consent as approved by the Ethical Committee of Hospital das Clínicas, Universidade Federal de Goiás, CAAE: 81316417.1.0000.5078, Brazil.
2.4. Genotyping, quality control and cytokine QTL mapping
DNA samples of 200 individuals were genotyped using commercially available SNP chip Illumina HumanOmniExpressExome-8 v1.0. Genotype calling was performed using Opticall 0.7.0 using default settings [32]. The imputation and quality control were performed as in the previous study [28]. In brief, samples with a call rate ≤ 0.99 were excluded, as were variants with Hardy-Weinberg equilibrium (HWE) ≤ 0.0001, and minor allele frequency (MAF) ≤ 0.1. Strands of variants were aligned and identified against the 1000 Genome reference panel using Genotype Harmonizer. In total there were 158 samples (Western ancestry) with both DNA and cytokine measurements.
Importantly, it is unclear if the functional consequences of the SNPs that the authors show are in any way correlated to PCM and its syndromes. Essentially, the SNPs identified are associated with a difference in immune function in vitro to Pb, but the pertinence of those findings to risk of PCM are not clear. If correlation to clinical disease is not possible, then at least consider complementing the findings with further functional studies, e.g. blocking IL1B or CD38 (separately) and determining the 2 in vitro/vivo effect on Pb infection.
Answer: We are grateful for these comments/ideas. As we stated before and added as Conclusions of the manuscript, we intend to do functional experiments to improve the understanding about the SNPs reported, since now we have the opportunity to evaluate some potential causal genes related to cytokine production induced by P. brasiliensis.
The correlation from PROM1 to CD133 to CD38 is not clear, and only becomes apparent in the Discussion (on page 16). These molecules need to be introduced and explained earlier, some (e.g. CD133) when they're first mentioned, and others (e.g. CD38) at least in the respective section in the Results.
Answer: In the present study we reported PROM1/CD133 as a potential causal gene controlling IL-22 production and CD38 with possible connection with PROM1, we have introduced these molecules in the Results (previous version). In addition, the text in the Discussion was adjusted accordingly.
3.2. Genome-wide mapping identifies genetic variations associated with innate- and adaptive-derived cytokines induced by P. brasiliensis
Third paragraph:
We also identified a SNP (rs62290169) in an intronic region of the PROM1 gene (at chromosome 4), which encodes the CD133 transmembrane protein. This protein has been used as a marker to isolate cancer stem cells in several tumors. Although it is highly expressed in embryonic stem cells its function in these cells remains unknown [37,38].
3.4. The PROM1 rs62290169 SNP regulates the surface expression of CD38 in T-helper lymphocytes
First paragraph:
The PROM1 gene is located in close proximity with the gene that encodes the CD38 molecule, which is a marker of T cell activation [41]. Furthermore three-dimensional chromatin structures, known as topologically associating domains (TADs), show that chromatin in the eukaryotic nucleus is divided into regions enriched in chromosomal contacts [42].
- Discussion - 8th paragraph:
Associated with IL-22 production, we identified a SNP (rs62290169) located in the PROM1 gene, which encodes the CD133 protein. Despite this protein is highly expressed in embryonic stem cells its function in these cells remains unknown [37,38]. The identified SNP is located in a non-coding and junction region with the gene encoding the CD38 molecule.
Here, we have discarded the last statement “which is a marker of T cell activation [41]”, to be not repetitive once it was explained in Results.
The implication of Raf1, but not Syk, by the use of molecular inhibitors is at risk for false interpretation if the inhibitors have bystander effects. This approach requires more proof of selective inhibition, or complementation with other techniques, or perhaps some evidence that the SNP in Dectin1 interferes with Raf1 interaction (although not a priori likely, given that it's only in the 3'UTR).
Answer: We agree with the referee that pharmacological inhibition is only one step to prove that one pathway was triggered; however, we have shown first that Dectin-1 pathway is triggered by P. brasiliensis antigens in PBMCs from healthy individuals (blocking the receptor with antibodies) and then the pharmacological inhibition was performed, which together confirmed Dectin-1 as relevant receptor to P. brasiliensis recognition and increase of IL-1b. We understand that what is missing here is the connection of these results with the SNP found. We intend to check this in further experiments.
Reviewer 2 Report
My comments have been properly addressed. I have not further criticisms.
Author Response

(The authors gave the same response as above.)
